# Molecular Mechanism of Calycosin Inhibited Vascular Calcification

**DOI:** 10.3390/nu16010099

**Published:** 2023-12-27

**Authors:** Zekun Zhou, Yi Li, Wei Jiang, Zengli Wang

**Affiliations:** 1College of Food Science and Nutritional Engineering, China Agricultural University, Beijing 100083, China; s20213061070@cau.edu.cn (Z.Z.); cauliyi@163.com (Y.L.); 2College of Biological Sciences, China Agricultural University, Beijing 100193, China; jiangwei01@cau.edu.cn

**Keywords:** calycosin, autophagy, lysosome, vascular calcification, SNARE complex

## Abstract

Vascular calcification (VC) is a pathological condition frequently observed in cardiovascular diseases. Primary factors contributing to VC are osteogenic differentiation of vascular smooth muscle and hydroxyapatite deposition. Targeted autophagy (a lysosome-mediated mechanism for degradation/recycling of unnecessary cellular components) is a useful approach for inhibiting VC and promoting vascular cell health. Calycosin has been shown to alleviate atherosclerosis by enhancing macrophage autophagy, but its therapeutic effect on VC has not been demonstrated. Using an in vitro model (rat thoracic aortic smooth muscle cell line A7r5), we demonstrated effective inhibition of VC using calycosin (the primary flavonoid component of astragalus), based on the enhancement of autophagic flux. Calycosin treatment activated AMPK/mTOR signaling to induce initiation of autophagy and restored mTORC1-dependent autophagosome–lysosome fusion in late-stage autophagy by promoting soluble N-ethylmaleimide-sensitive factor attachment protein receptor (SNARE) complex formation, thereby preventing stoppage of autophagy in calcified cells. Calycosin substantially reduced degrees of both osteogenic differentiation and calcium deposition in our VC cell model by enhancing autophagy. The present findings clarify the mechanism whereby calycosin mitigates autophagy stoppage in calcified smooth muscle cells and provide a basis for effective VC treatment via autophagy enhancement.

## 1. Introduction

Vascular calcification (VC) is a pathological condition frequently observed in patients with atherosclerosis, chronic kidney disease, diabetes mellitus, and other chronic diseases [1,2,3]. Because of its negative effect on vascular compliance and vasorelaxation, VC is strongly associated with adverse cardiovascular outcomes and increased mortality [4]. It results primarily from the activity of vascular smooth muscle cells (SMCs), which undergo a phenotypic shift that plays a key role in this process [1]. In a calcifying environment, SMCs are transformed into osteoblasts, resulting in the loss of contractile markers such as smooth muscle α-actinin and smooth muscle 22 alpha protein (SM22α) [5]. Conversely, the expression of osteogenic markers such as osteopontin (OPN), RUNT-related transcription factor 2 (RUNX2), and bone morphogenetic protein 2 (BMP2) is upregulated in a calcifying environment [6]. The release of hydroxyapatite-rich matrix vesicles by differentiated SMCs results in the deposition of calcium and phosphate ions in arterial linings [7].

The pathogenesis of VC involves the dysregulation of autophagy, a lysosome-mediated mechanism leading to the orderly degradation and recycling of unnecessary or dysfunctional cellular components. The fundamental component of VC is hydroxyapatite, Ca_10_(PO_4_)_6_(OH)_2_ [8]. Administration of nano-sized hydroxyapatite interferes with normal lysosome function, thereby inhibiting autophagic flux [9]. Such disruption leads to the accumulation of autophagosomes and autolysosomes, which are subsequently converted into Ca^2+^-bearing exosomes [10]. This process accelerates the calcification of the extracellular matrix. Impairment of autophagy promotes osteogenic differentiation of SMCs. Thus, autophagy plays a major role in the process of arterial calcification. Induction of autophagy was shown to have a mitigating effect on the osteogenic differentiation of vascular SMCs, with consequent reduction of VC [11,12]. We therefore suspect that therapeutic approaches designed to induce autophagy may effectively mitigate SMC differentiation and inhibit VC.

Mammalian target of rapamycin complex-1 (mTORC1), one of the two structurally and functionally distinct protein complexes formed by mammalian target of rapamycin (mTOR), plays a key role in the control of multiple cell physiological processes, including metabolism, protein synthesis, proliferation, and autophagy [13,14]. It is involved in the initiation of autophagy and it inhibits the autophagosome–lysosome fusion process. mTORC1 is located downstream of AMP-activated protein kinase (AMPK). Inhibition of mTORC1 mitigates the phosphorylation of Ser757 on autophagy-activated kinase 1 (ULK1), the downstream target of AMPK, upon AMPK activation. AMPK concurrently phosphorylates Ser317 and Ser777 on ULK1, thus initiating autophagy [15]. mTORC1 also directly induces phosphorylation in the protein coding gene VAMP8 (vesicle associated membrane protein 8), which leads to the disruption of SNARE complex formation, the consequent blocking of autophagosome–lysosome fusion, and the inhibition of autophagic flux [16]. Activation/inhibition of mTOR are therefore key factors in the regulation of autophagy-related pathways.

Flavonoids (polyphenolic secondary metabolites from plants) are useful in the prevention and treatment of cardiovascular diseases. Flavonoid consumption has been correlated with a reduction in cardiovascular disease-related death rates [17]. The herb astragalus (*Astragalus membranaceus*; family Fabaceae), known for centuries in traditional Chinese medicine, contains flavonoids and is used in this way. Treatment with astragalus extracts was found to inhibit the calcification of SMCs susceptible to hyperphosphate-induced calcification [18]. Calycosin, the primary flavonoid component of astragalus, reduced the severity of atherosclerosis in an ApoE gene knockout mouse model, based on the enhancement of macrophage autophagy. Calycosin also displays antioxidant and anti-inflammatory effects [19]. The detailed roles of calycosin in regulation of autophagy in SMCs and in VC remain unclear. We used an in vitro A7r5 cell model to evaluate the effects of calycosin on autophagosome–lysosome fusion, and its mitigating effect on osteogenic differentiation and calcium accumulation in calcified SMCs. We observed that calycosin: (i) activated the AMPK/mTOR signaling pathway, thereby initiating autophagy; and (ii) enhanced the formation of the mTORC1-mediated SNARE complex, which includes syntaxin 17 (STX17), synaptosomal-associated protein 29 (SNAP29), and VAMP8. Our findings clarify the molecular pathways underlying the therapeutic effects of calycosin on calcified SMCs in vitro.

## 2. Materials and Methods

### 2.1. Cell Culture and Lentiviral Transduction

Rat thoracic aortic SMC line A7r5 was obtained from the Chinese Academy of Sciences Cell Bank (Shanghai, China) and grown in DMEM with 10% FBS in 5% CO_2_ atmosphere at 37 °C. All experiments used cells grown for 3–8 generations.

Lentiviral shRNA for VAMP8 silencing was from GeneCopoeia (Guangzhou, China). Lipo8000 Transfection Reagent (Beyotime Biotechnology Co.; Shanghai, China) was used to treat A7r5 cells with nontargeted (NC shRNA) or targeted (VAMP8 shRNA) shRNA for 48 h, incubated with puromycin (2.5 μg/mL) for 1 week to form stable cell line, and then the cells were used for the following experiments.

For calycosin experiments, cells after 80% confluence were cultured in growth medium with various calycosin concentrations for 24 h, then transferred to DMEM with 10% FBS and 10 mM β-glycerophosphate (β-GP) for 5 d. Cells were grown in DMEM with 10% FBS as uncalcified control. Medium was replaced every other day. 

### 2.2. Reagents and Antibodies

Calycosin (cat #B20846), bafilomycin A1 (Baf A1; cat #S17106), rapamycin (Rapa; cat #T54160), and MHY1485 (cat #S31626) were from Yuanye Bio-Technology Co. (Shanghai, China). β-glycerophosphate (cat #G9422) was from Sigma-Aldrich (Shanghai, China). LysoTracker Deep Red was from Beyotime. Self-quenched BODIPY FL conjugate of BSA (DQ-BSA; green; cat #ab286868) was from BioVision, Inc. (Milpitas, CA, USA). Antibodies against RUNX2, BMP2, OPN, Cathepsin B (CTSB), microtubule associated protein 1 light chain 3 beta 2 (LC3B), sequestosome 1 (p62), AMPK, p-AMPK (Thr172), mTOR, *p*-mTOR, ULK1, p-ULK1 (757), VAMP8, and SNAP29 were from Abcam (Cambridge, UK). Antibodies against SM22α, lysosome-associated membrane protein 2 (LAMP), p-S6K, and ribosomal protein S6 kinase (S6K) were from Cell Signaling Technology (Danvers, MA, USA). Antibodies against β-tubulin, β-actin, GAPDH, and STX17 were from Sino Biological, Inc. (Beijing, China).

### 2.3. Calcium Content and Alkaline Phosphatase (ALP) Activity

Calcified A7r5 cells were washed twice with PBS and lysed with RIPA lysis buffer (Solarbio Science & Technology Co.; Beijing, China), and calcium content and ALP activity expression of supernatant were measured using respective assay kits (cat #s S1063S and P0321S; Beyotime). Protein level of supernatant was measured using BCA (bicinchoninic acid) assay kit (cat #P0010S; Beyotime).

### 2.4. Alizarin Red S Staining

For detection of calcification, cells were rinsed with PBS, fixed in 4% paraformaldehyde solution, treated with alizarin red S solution (Solarbio) for 20 min at 37 °C, washed with PBS, and photographed.

### 2.5. Protein Extraction and Western Blotting

Cells were lysed with RIPA lysis buffer as above, and proteins in lysates were separated by SDS-PAGE and transferred to nitrocellulose or PVDF membranes. Membranes were sealed for 1 h with 5% skim milk in PBST at room temperature and incubated overnight at 4 °C with primary antibody and then for 1 h at room temperature with HRP-tagged secondary antibody. Protein bands were visualized using ECL substrate.

### 2.6. Immunoprecipitation

Cells were lysed with immunoprecipitation (IP)/protein lysate (cat #P0013; Beyotime; including protease inhibitor combination) for 30 min. Lysates were incubated with anti-VAMP8 or anti-IgG antibodies at 4 °C for 24 h, incubated with protein G magnetic beads (Cell Signaling Technology) for 1.5 h, washed 4× with IP/protein immunoblotting lysate, and subjected to western blotting analysis.

### 2.7. Autophagic Flux Assay

Lentiviruses used were RFP-GFP-LC3B (cat #HB-LP210; Han Biomedical, Inc., Taiwan) and Ad-GFP-LC3B (cat #C3006; Beyotime). A7r5 cells (4 × 10^5^/well) were transfected with lentivirus for 48 h, incubated with puromycin (2.5 μg/mL) for 1 week to form stable cell line [20], and treated as described in Section 2.1. Autophagic flux was assayed using flow cytometric analysis using FACSCalibur platform (BD Biosciences; Franklin Lakes, NJ, USA).

### 2.8. Bovine Serum Albumin (BSA) Dequenching Assay

Degree of lysosomal protein breakdown (degradation) was assayed using DQ-BSA labeling method [21]. Cells in 6-well plates were treated with calycosin (0–20 μM) for 24 h, followed by another 4 h after either adding or not adding Baf A1 (100 nM) or rapamycin (200 nM). After that the medium was changed and added to β-GP (10 mM) for 5 d, then incubated with DQ-BSA (BioVision; 10 μg/mL) for 1 h at 37 °C and washed twice with PBS. DQ-BSA fluorescence was assayed using flow cytometry on BD FACSCalibur platform.

### 2.9. LysoTracker Assay

Intralysosomal acidity was measured by LysoTracker Deep Red method. Cells were treated with calycosin (0–20 μM) for 24 h, followed by another 4 h after either adding or not adding Baf A1 (100 nM) or rapamycin (200 nM). After that the medium was changed and added to β-GP (10 mM) for 5 d, then incubated in 6-well plates with LysoTracker Crimson (100 nM; cat #C1046; Beyotime) for 30 min at 37 °C, and washed with PBS. Fluorescence intensity was assayed imme-diately using BD FACS Calibur platform.

### 2.10. Statistical Analyses

Statistical analyses were performed using SPSS 26.0 software program (SPSS; Chicago, IL, USA), and data presented as mean ± SD. Differences among multiple groups were analyzed by one-way ANOVA, with *p* < 0.05 as criterion for statistical significance.

## 3. Results

### 3.1. Calycosin Treatment Inhibits β-GP-Induced Osteogenic Differentiation and Calcification in A7r5 Cells In Vitro

In order to evaluate the effect of calycosin on the calcification level of SMCs in vitro, we used a calcification model based on cultured A7r5 cells treated with β-glycerophosphate (β-GP). Alizarin red S staining revealed that the quantity of calcium nodules was much higher in β-GP-treated cells than in controls. The presence of calycosin inhibited such calcification (Figure 1A). β-GP treatment resulted in increased levels of indicators associated with the calcification process (calcium content, ALP activity), but such increases were counteracted by the presence of calycosin (Figure 1B). β-GP treatment led to elevated expression levels of osteogenic gene markers (BMP2, RUNX2, OPN) (Figure 1C,D). On the other hand, the level of SM22α, a contraction marker specific to A7r5 cells, declined. Calycosin inhibited osteoblast differentiation and inhibited the upregulation of Runx2, BMP2, and OPN as well as the downregulation of SM22α in a dose-dependent manner. These findings, taken together, indicate that calycosin inhibits the calcification of SMCs in vitro.

### 3.2. Calycosin Increases Autophagic Flux, but Has No Effect on Lysosome Function in Calcified Cells

Autophagy, by inhibiting osteogenic differentiation of SMCs, inhibits VC [22]. We evaluated the effect of calycosin on autophagy by examining the conversion of LC3B-I to LC3B-II and the presence of autophagic puncta in A7r5 cells stably expressing GFP-LC3B. LC3B-II accumulation in A7r5 cells was increased via β-GP treatment, and further increased via the addition of calycosin (Figure 2A,B). LC3B-II levels can be controlled either via upregulation during early-stage autophagy or downregulation during late-stage autophagy. Autophagosome–lysosome fusion was inhibited by Baf A1, leading to increased LC3B-II and p62 levels (Figure 2C,D). In GFP-LC3B-transfected cells, LC3B-II expression was strongly increased via combined treatment with calycosin and Baf A1. The number of autophagic puncta was increased via calycosin treatment (Figure 2E), and more strongly increased via combined calycosin/Baf A1 treatment than via Baf A1 alone. These findings confirm the enhancing effect of calycosin on autophagosome production. 

The β-GP-induced accumulation of autophagy-specific substrate p62 was inhibited by calycosin in a dose-dependent manner, and this effect was reversed by Baf A1 treatment, suggesting the ability of calycosin to mitigate disruption of autophagic flux in the presence of β-GP. This possibility was evaluated using experiments with RFP-GFP-LC3B lentivirus transfected cells. In the treated cells, relative to controls, the presence of β-GP resulted in a ~32% reduction of autophagic lysosomes (RFP+GFP-), whereas the presence of calycosin had no such effect (Figure 2F,G). The effect of calycosin on autophagic lysosomes was reversed in the Baf A1-treated group. These findings, taken together, demonstrate that calycosin effectively mitigated the damage inflicted by β-GP on autophagic flux.

The restoration of autophagic flux may result from an enhanced degree of autophagosome–lysosome fusion or lysosomal degradation ability. We evaluated lysosome function using several techniques. Lysosomal degradation ability was investigated using a BSA dequenching assay test (Section 2.8). DQ-BSA fluorescence intensity was not notably altered by β-GP or calycosin treatment, indicating that calycosin does not affect lysosomal degradation ability (Figure 3A). β-GP treatment strongly increased levels of mature tissue protease B (CTSB; 2.5-fold increase), a lysosomal enzyme involved in autophagy, and of lysosome-associated membrane protein 2 (LAMP2; 2-fold increase), a lysosome marker (Figure 3C–F). Calycosin treatment greatly reduced both CTSB and LAMP2 levels. The level of acidic lysosomes was assayed using LysoTracker Red (LTR). LTR fluorescence intensity was enhanced by 26% via β-GP treatment but was reduced to near-control level via 20 μM calycosin treatment (Figure 3B); i.e., β-GP-induced increase of lysosome count was counteracted by calycosin. Calycosin effectively inhibited aberrant accumulation of lysosomes, although it had no effect on lysosome function. The findings shown in Figure 2G are consistent with the observed effects of β-GP and calycosin treatments on the quantity of autophagic lysosomes. In conclusion, calycosin has the ability to mitigate disruption of autophagic flux by promoting autophagosome–lysosome fusion, even though it does not affect lysosome function.

### 3.3. Calycosin Initiates Autophagy via Activation of AMPK/mTOR Signaling Pathway in Calcified Cells

mTORC1 functions as a serine/threonine protein kinase and plays an essential role as a negative regulator in both early and late stages of autophagy. Control of autophagy is mediated by AMPK, which inhibits mTOR phosphorylation (see Introduction). The β-GP-induced phosphorylation of mTOR (Ser2448) was inhibited, in a dose-dependent manner, by calycosin treatment (Figure 4A). The total mTOR level remained constant. ULK1 (Ser757) levels increased, while p-S6K (Ser371) levels decreased in a dose-dependent manner, suggesting a reduction in mTORC1 activity. The phosphorylation level of AMPK was restored via calycosin treatment, indicating the ability of calycosin to reverse the β-GP-induced inhibition of AMPK/mTOR signaling (Figure 4A,B). 

In view of the above findings, we examined the possible involvement of the AMPK/mTOR signaling pathway in the regulatory effects of calycosin on autophagy initiation. The effects of MHY1485 (a cell-permeable mTOR activator) on autophagy in the presence of calycosin were evaluated. MHY1485 counteracted the inhibitory effect of calycosin on mTOR phosphorylation and its facilitation of LC3B-II phosphorylation, indicating that calycosin activates autophagy by inhibiting mTOR phosphorylation (Figure 4C,D). Combined treatment using MHY1485/calycosin resulted in the downregulation of the inhibitory effect of calycosin on mTOR phosphorylation and its promoting effect on autophagy initiation. GFP-LC3B-overexpressing A7r5 cells were studied in order to clarify the role of mTOR in calycosin-induced autophagy. The cells were incubated with calycosin alone or MHY1485/calycosin and subsequently with β-GP. The number of LC3B puncta strongly increased after calycosin treatment, and this increase was reversed by MHY1485 (Figure 4E). These findings indicate that calycosin induces autophagy initiation through the activation of the AMPK/mTOR signaling pathway.

### 3.4. Calycosin Inhibits Calcification by Promoting SNARE Complex-Mediated Autophagosome–Lysosome Fusion

mTORC1 plays a role in VAMP8 phosphorylation, resulting in the inhibition of autophagosome–lysosome fusion through the inhibition of SNARE complex formation, which involves syntaxin 17 (STX17), synaptosomal-associated protein 29 (SNAP29), and vesicle associated membrane protein 8 (VAMP8) [16]. We examined expression levels of SNARE complex proteins under β-GP and calycosin treatments. STX17, SNAP29, and VAMP8 levels were strongly reduced via β-GP treatment, but increased in a dose-dependent manner via calycosin treatment (Figure 5A,B). To evaluate the involvement of the mTOR-mediated inhibition of SNARE formation in the calycosin-induced enhancement of autophagosome–lysosome fusion, we examined expression levels of STX17, SNAP29, and VAMP8 following mTOR activation. MHY1485 reversed the upregulation of STX17, SNAP29, VAMP8 and the downregulation of p62 caused by calycosin. (Figure 5C,D). Observed alterations in signaling by RFP+GFP- (autolysosomes) indicated that calycosin effectively counteracted the β-GP-induced disruption of autophagic flux (Figure 5E,F); however, MHY1485 inhibited the effect of calycosin. β-GP treatment disrupted interactions among STX17, SNAP29, and VAMP8, but such disruption was counteracted by calycosin treatment (Figure 6A,B). These findings, taken together, indicate that calycosin enhances autophagosome–lysosome fusion by preventing disruption of the STX17-SNAP29-VAMP8 complex.

To evaluate the role of VAMP8 in calycosin-enhanced autophagosome–lysosome fusion, we generated an A7r5 strain in which VAMP8 expression was silenced. VAMP8 silencing inhibited the calycosin-induced reduction of p62 (Figure 6C,D). LC3B-II accumulation was greater in the group with combined VAMP8/calycosin treatment vs. calycosin treatment alone. In VAMP8-silenced cells treated with calycosin, the level of autophagic lysosomes (RFP+GFP-) was notably reduced (Figure 6E). β-GP treatment of parental A7r5 cells led to the inhibition of autophagosome–lysosome fusion. Such inhibition was strongly mitigated by calycosin treatment but exacerbated by VAMP8 silencing (Figure 6F). Calycosin promotes autophagosome–lysosome fusion by upregulating VAMP8. 

The mitigating effects of calycosin on calcified A7r5 cells (e.g., reduction of calcium nodule number, calcium content, and ALP activity) were reversed by VAMP8 silencing (Figure 7A,B). We found that VAMP8 silencing enhanced the ability of β-GP to induce calcification and reversed the ability of calycosin to mitigate β-GP-induced osteogenic differentiation. This reversal was evident when comparing the effects of calycosin treatment alone vs. combined treatment with calycosin and VAMP8 silencing. VAMP8 silencing resulted in increased expression of key osteogenic gene markers (BMP2, RUNX2, OPN). VAMP8 silencing also reduced levels of the A7r5-specific contraction marker SM22α (Figure 7C,D). Our findings, taken together, indicate that the ability of calycosin to alleviate the calcification of A7r5 cells via autophagy is achieved by promoting SNARE complex formation.

## 4. Discussion

Vascular calcification (VC), a pathological condition frequently observed in chronic cardiovascular diseases, typically occurs in the intimal and medial layers of the arterial wall or in valves [1]. VC is characterized histopathologically by calcium phosphate (Ca_3_(PO_4_)_2_) accumulation, and by osteogenic differentiation of SMCs [23]. Calcium deposition and endothelial dysfunction may trigger an inflammatory response, leading in severe cases to cellular necrosis and blood clot formation. The process of autophagy inhibits osteogenic differentiation of SMCs by reducing the phenotypic transformation of these cells. Such transformation involves the downregulation of factors associated with contractile phenotypes (smooth muscle α-actinin, SM22α) and upregulation of calcification markers (ALP, RUNX2, BMP, OPN) [6,24]. The function of autophagic lysosomes plays an essential role in the inhibition of VC [25]. Recent studies demonstrate the ability of calycosin to mitigate cardiovascular disorders by activating autophagy via several pathways. Details of the mechanisms whereby calycosin enhances autophagy in SMCs, inhibits VC, and mitigates cardiovascular diseases remain unclear [19,26,27]. The results of the present study demonstrate that calycosin reduces β-GP-induced calcification and autophagic flux damage in rat thoracic aortic SMC (A7r5). Previous studies have shown that the ability of calycosin to inhibit osteogenic differentiation of SMCs and in vitro calcium deposition clearly involves the process of autophagy.

Research interest in autophagy has increased substantially during the past decade, in view of its wide-ranging regulatory functions in cardiovascular diseases [28,29]. Several drugs that specifically target autophagy have been developed for the potential treatment of VC. Metformin, for example, inhibits VC by promoting autophagy through the activation of the AMPK/autophagy-related 3 (Atg3) pathway, leading to the degradation of RUNX2 via p62-mediated mechanisms [30]. Irisin prevents VC in chronic kidney disease by inducing autophagy and inhibiting NLRP3-mediated pyroptosis in vascular SMCs [31]. We observed in the present study that calycosin affects cellular calcification through two distinct mechanisms: (i) initiating early-stage autophagy; and (ii) promoting autophagosome–lysosome fusion during late-stage autophagy. Mitigation of calcification is clearly based on these effects. 

We further studied the mechanism by which calycosin regulates autophagy. AMPK exerts an active effect in early-stage autophagy through ULK1 phosphorylation. The mTOR pathway has a negative regulatory effect during early-stage autophagy and facilitates VAMP8 phosphorylation during late-stage autophagy. VAMP8 phosphorylation disrupts the assembly of the STX17-SNAP29-VAMP8 complex, thereby inhibiting autophagosome–lysosome fusion [16]. L.-Q. Yuan’s group reported that the activation of the AMPK/mTOR signaling pathway led to induction of autophagy in calcified SMCs [32]. Our findings indicate a comparable effect of calycosin on AMPK/mTOR signaling in A7r5 cells, which facilitates early-stage autophagy. We used a BSA dequenching assay and LysoTracker analysis to examine the effects of calycosin treatment on lysosomal function and autophagosome–lysosome fusion. Calycosin treatment had no effect on lysosome number or lysosomal degradation ability. β-GP treatment disrupted the formation of the STX17-SNAP29-VAMP8 complex, an mTORC1-mediated process essential for autophagosome–lysosome fusion. The autophagy process was blocked by such disruption, but was restored by calycosin treatment. The effects of calycosin on autophagy initiation and autophagosome–lysosome fusion promotion were blocked by the autophagy inhibitor Baf A1 as well as the mTOR activator MHY1485. VAMP8 silencing in A7r5 cells completely inhibited the alleviation of calcification by calycosin. Thus, calycosin inhibits the calcification of A7r5 by promoting autophagy through the activation of the AMPK/mTOR signaling pathway and by maintaining autophagosome–chromosome–lysosome fusion through the promotion of STX17-SNAP29-VAMP8 complex formation.

## 5. Conclusions

In conclusion, the observed mitigating effect of calycosin on SMC calcification is based on its activation in the AMPK/mTOR signaling pathway and promotion of STX17-SNAP29-VAMP8 complex formation. This study clarifies the molecular mechanisms underlying the therapeutic effects of calycosin on VC. Our findings demonstrate the inhibitory role of the STX17-SNAP29-VAMP8 complex in SMC calcification and suggest potential molecular targets and pharmacological drugs for the prevention and treatment of VC.

## Figures and Tables

**Figure 1 nutrients-16-00099-f001:**
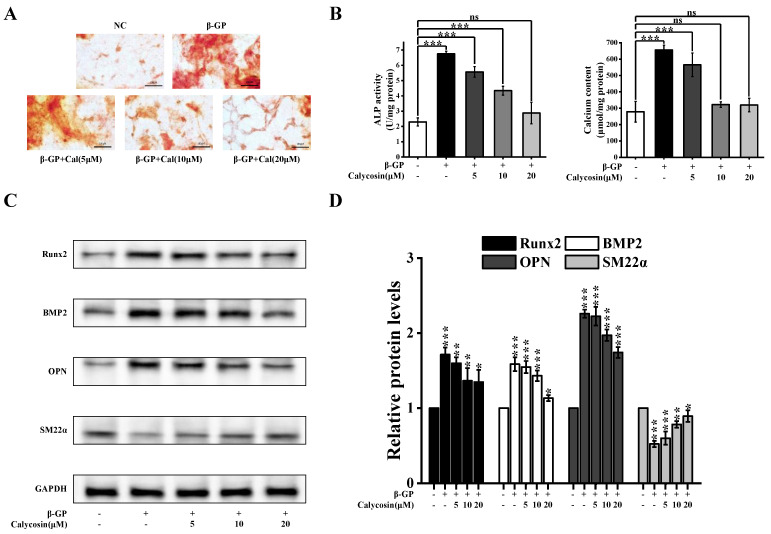
Calycosin mitigates β-GP-induced VC in vitro. A7r5 cells were pretreated with calycosin (0–20 μM) for 24 h, medium was added to β-GP (10 mM), and incubation continued for 5 d. (**A**) Alizarin red S staining to reveal calcium nodules (*n* = 6 biological replicates). (**B**) ALP activity and calcium content (*n* = 6 biological replicates). (**C**,**D**) RUNX2, BMP2, OPN, and SM22α expression levels assayed using western blotting (*n* = 6 biological replicates). Data are presented as mean ± SD. All the treatment groups were compared with the blank control group. * 0.01 < *p* < 0.05; ** 0.001 < *p* < 0.01; *** *p* < 0.001; ns represents no significant difference.

**Figure 2 nutrients-16-00099-f002:**
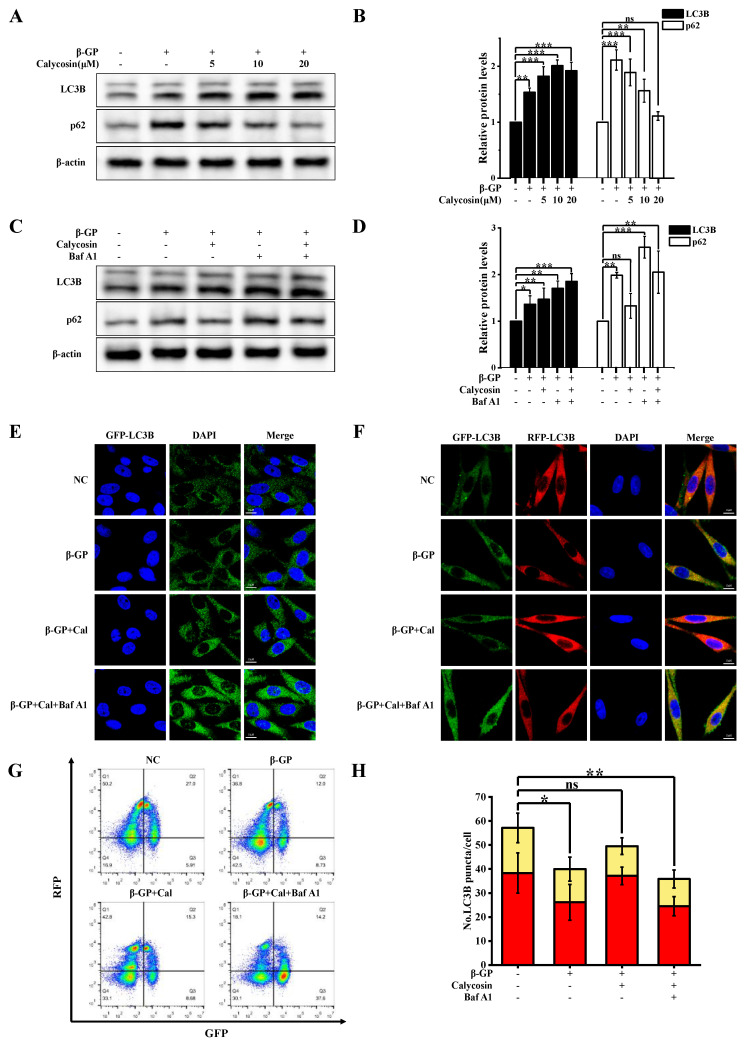
Calycosin mitigates β-GP-induced impairment of impaired autophagy. Cells were incubated with calycosin (0–20 μM) for 24 h, then for another 4 h after adding or not adding 100 nM Baf A1; then, the medium was changed and added to β-GP (10 mM) for 5 d. (**A**,**B**) p62 and LC3B protein levels assayed using western blotting (*n* = 6 biological replicates). (**C**,**D**) Cell lysates were western blotted with anti-LC3B or anti-p62 antibody (*n* = 6 biological replicates, calycosin = 20 μM). (**E**) Representative images of cells transfected with GFP-LC3B and then treated as in (**B**) (calycosin = 20 μM). (**F**) Representative images of cells expressing tandem RFP-GFP-tagged LC3B, treated as in (**B**). Scale bars: 10 μm (*n* = 6 biological replicates, calycosin = 20 μM). (**G**,**H**) Cells stably expressing RFP-GFP-LC3B were treated as in (**B**), and autophagy index (ratio of area of RFP+GFP-dots (autophagic lysosomes) to area of RFP+GFP+dots (autophagosomes)) was determined using flow cytometry (*n* = 6 biological replicates, calycosin = 20 μM). Statistical notations as in Figure 1.

**Figure 3 nutrients-16-00099-f003:**
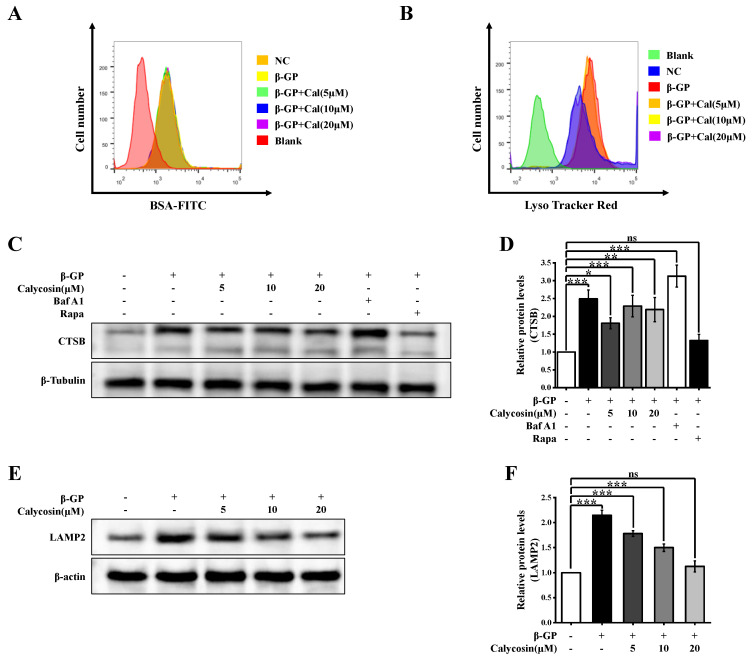
Calycosin affects autophagosome–lysosome fusion, but not lysosomal tissue protease maturation or lysosomal pH. Cells were incubated with calycosin (0–20 μM) for 24 h, then for another 4 h after either adding or not adding Baf A1 (100 nM) or rapamycin (200 nM). Then the medium was changed and added to β-GP (10 mM) for 5 d. (**A**) Cells were stained with DQ-BSA (10 μg/mL) for 1 h, and DQ-BSA fluorescence intensity was quantified by flow cytometric analysis (*n* = 6 biological replicates). (**B**) Effects of calycosin and β-GP treatment on mean fluorescence intensity of cells stained with LysoTracker Red (*n* = 6 biological replicates). (**C**,**D**) Cells were subjected to cathepsin B (CTSB) immunoblotting analysis (*n* = 6 biological replicates). (**E**,**F**) Cells were subjected to LAMP2 immunoblotting analysis (*n* = 6 biological replicates). Statistical notations as in Figure 1.

**Figure 4 nutrients-16-00099-f004:**
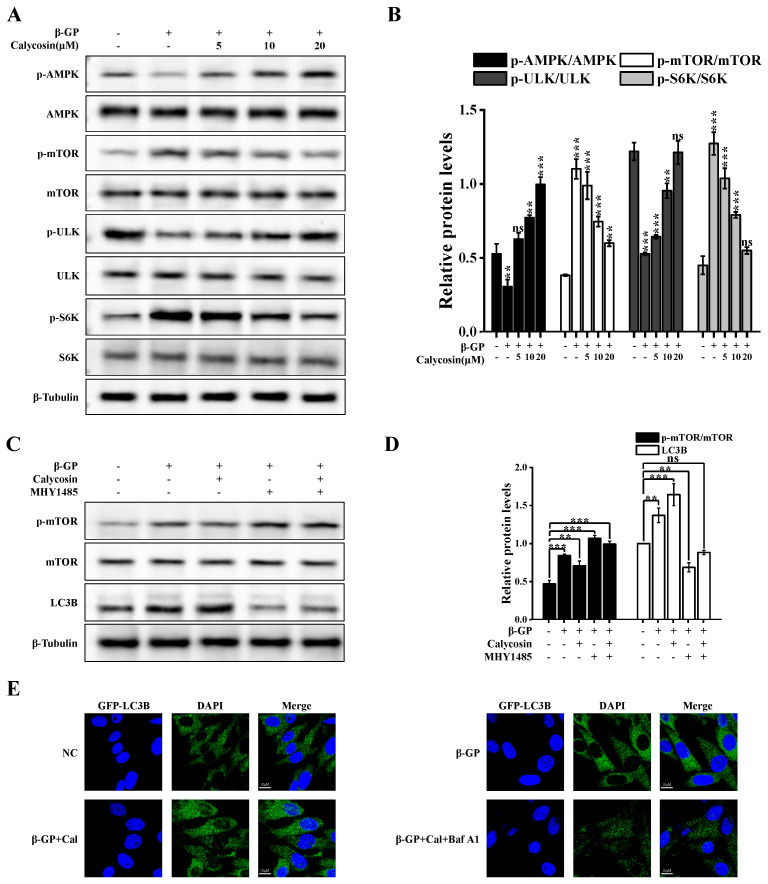
Calycosin initiates autophagy through AMPK/mTOR signaling. Cells were incubated with calycosin (0–20 μM) for 24 h, then for another 4 h after either adding or not adding MHY1485 (200 nM); then, the medium was changed and added to β-GP (10 mM) for 5 d. (**A**,**B**) Immunoblotting analysis of AMPK, mTOR, ULK1, and S6K levels, and p-AMPK (Thr172), p-mTOR (Ser2448), p-ULK1 (757), and p-S6K (Thr389) phosphorylation levels, in cells treated with indicated concentrations of calycosin (*n* = 6 biological replicates). (**C**,**D**) LC3B, p-mTOR (Ser2448), and mTOR expression levels in cells either treated or not treated with calycosin or MHY1485 (*n* = 6 biological replicates, calycosin = 20 μM). (**E**) Representative images of cells transfected with GFP-LC3B for 48 h. Scale bar: 10 μm (*n* = 6 biological replicates, calycosin = 20 μM). Statistical notations as in Figure 1.

**Figure 5 nutrients-16-00099-f005:**
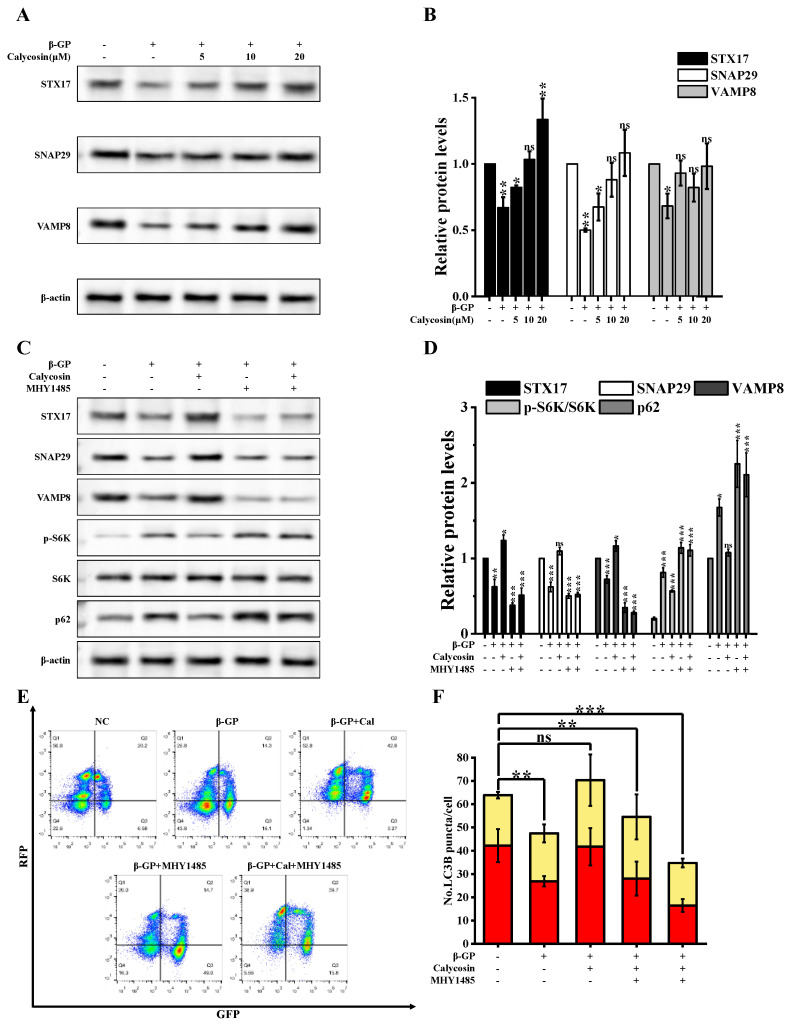
Calycosin counteracts disruption of SNARE complex formation by β-GP. Cells were incubated with calycosin (0–20 μM) for 24 h, then for another 4 h after either adding or not adding MHY1485 (200 nM); then, the medium was changed and added to β-GP (10 mM) for 5 d. (**A**,**B**) STX17, SNAP29, and VAMP8 protein expression levels were assayed using western blotting (*n* = 6 biological replicates). (**C**,**D**) Levels of SNARE complex proteins and mTOR substrates were determined using immunoblotting analysis (*n* = 6 biological replicates, calycosin = 20 μM). (**E**,**F**) Cells were transfected with RFP-GFP-LC3B lentivirus for 48 h, treated as in (**C**), and autophagy index was quantified using flow cytometric analysis (*n* = 6 biological replicates, calycosin = 20 μM). Statistical notations as in Figure 1.

**Figure 6 nutrients-16-00099-f006:**
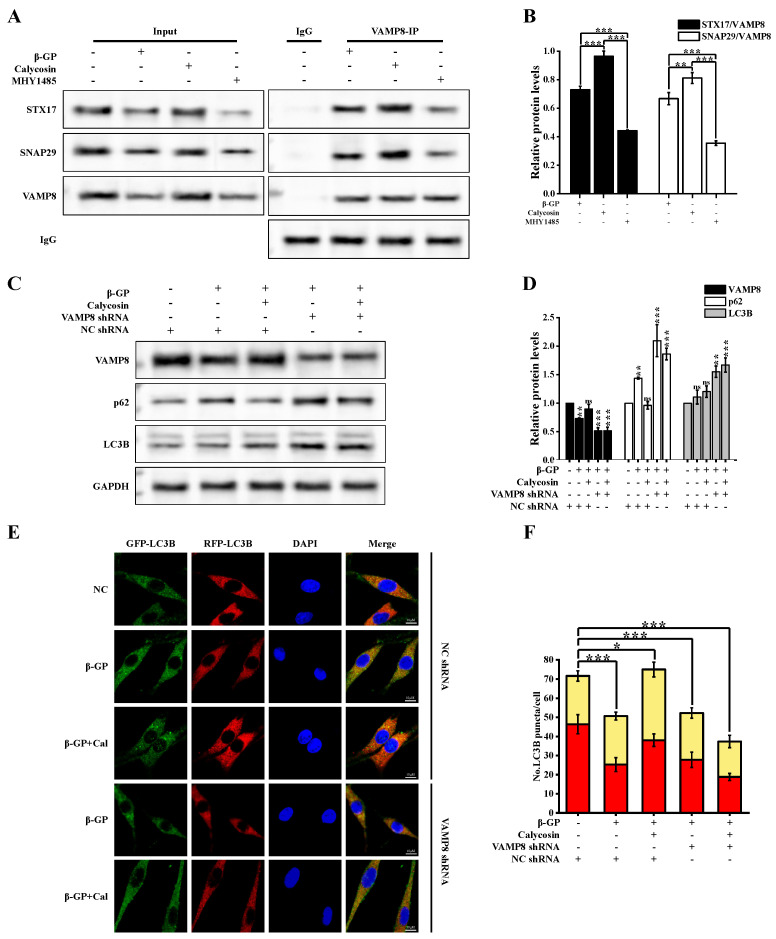
VAMP8 mediates autophagy and calycosin-induced calcification. (**A**,**B**) Cells were incubated with calycosin (20 μM) for 24 h, then for another 4 h after either adding or not adding MHY1485 (200 nM); then, the medium was changed and added to β-GP (10 mM) for 5 d, and interaction of VAMP8 with SNAP29/STX17 was evaluated using IP analysis. VAMP8, SNAP29, and STX17 expression levels were determined using respective antibodies (*n* = 6 biological replicates). (**C**,**D**) Cells were treated as described in Section 2.1, and VAMP8, LC3B, and p62 expression was assayed using western blotting (*n* = 6 biological replicates, calycosin = 20 μM). (**E**) Cells were transfected with RFP-GFP-LC3B lentivirus as described in Section 2.7, then treated as described in Section 2.1 and subjected to LC3B puncta immunofluorescence analysis. Scale bar: 10 μm (*n* = 6 biological replicates, calycosin = 20 μM). (**F**) Cells were treated as in (**E**), and autophagy index was determined using flow cytometric analysis (*n* = 6 biological replicates). Statistical notations as in Figure 1.

**Figure 7 nutrients-16-00099-f007:**
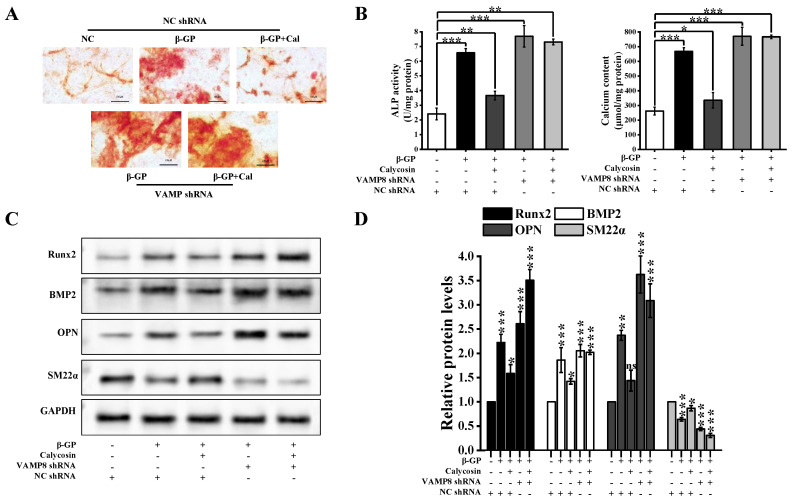
VAMP8 mediates the inhibitory effect of calycosin on calcification. Cells were treated as described in Section 2.1 (calycosin = 20 μM). (**A**) Alizarin red S staining to reveal calcium nodules (*n* = 6 biological replicates). (**B**) ALP activity and calcium content (*n* = 6 biological replicates). (**C**,**D**) RUNX2, BMP2, OPN, and SM22α expression levels assayed using western blotting (*n* = 6 biological replicates). Statistical notations as in Figure 1.

## Data Availability

Data are contained within the article.

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
