# Peer review of "Molecular Mechanism of Calycosin Inhibited Vascular Calcification"

_nutrients, 2023, doi:10.3390/nu16010099_

Round 1
Reviewer 1 Report
Comments and Suggestions for Authors
The study demonstrated how calycosin (the primary flavonoid component of astragalus) can inhibit the VSMC calcification in molecular aspects, which is enhancement of autophagic flux, such as through activating AMPK/mTOR signaling to induce initiation of autophagy, and restoring mTORC1-dependent autophagosome-lysosome fusion by promoting SNARE complex formation.
The approach of their study rationale and the findings are novel and extend the knowledge of a particular functional component could the effective potential therapeutic component for atherosclerosis treatment.
Reviewer 2 Report
Comments and Suggestions for Authors
In this article Zhou et al provide evidence that calycosin, a flavonoid from Astraglus, recapitulates the anti-calcific effects of Astralagus extracts on VSMC, indicating that it may be the active compound. Previously, calycosin was shown to reduce atherosclerosis in ApoE mice through enhanced macrophage autophagy. Overall, the data supports the hypothesis that calycosin can have further pleiotropic effects in atherosclerosis through inhibition of cellular calcification of VSMC in an in vitro model using A7r5. In itself this is significant from the translational perspective since drugs with pleiotropic effects on different cell types (provided they all go "in the same direction") are emerging as better suited for treatment of complex diseases. The authors use well-stablished and accepted approaches (from calcification protocols to methods to assess autophagic flow).
In spite of that overall enthusiasm, there are some conceptual shortcomings: a) experiments are prevention (pretreatment of A7r5). For significance in clinical intervention, reversion (difficult to set up) or, more realistically, 'slow down' (easy to set up) should be provided.
b) no evidence that calycosin as such in vivo (prevention/reversion) may reduce the calcification of atherosclerotic plaques (if it was published before, it should be mentioned)
Experimental design:
a) It is at times impossible to assess the time points or doses.
b) Some stats are comparing the wrong groups, or worse, the groups being compared are not even indicated. Example fig 3D and 5D; stats needed in 6F.
c) In most figures, it is impossible to understand the timelines or doses of calycosin. which is the dose in Fig 2-4? Or, more complicated to understand, are the signaling studies in figure 4 done at day 5 or in acute exposures?
d) in fig 6, is the shRNA stable transfection? or transient? how does this relate to the 'undisclosed' time of the experimental end point? (5 days? a few hours?) Is the transfection done at day 3 of the 5 days protocol with the B-GP???
All of these shortcomings make virtually impossible to evaluate the 'quality' of the data. The authors should provide a careful revision of the way they present their information and how that affects (or not) their initial data interpretation in this version.
Other: please clarify the meaning of (>xx.x%) by each of the reagents. Write for a wide audience. Do not take shortcuts with the statistical annotations. They may not all be as in Fig 1 -maybe they are for the meaning of the asterisks but were they handled the same way? is the data always normal?
Comments on the Quality of English LanguageAuthors need to revise the syntax, grammar and semantics. It appears at times as if a translator (AI or human) may have been used (examples: LC3 sites or dots, instead of "puncta"; combination treatment for "combined treatment"; protein blotting, negating the accepted Western blotting original name; suppressed and inhibition are not interchangeable; functioning for function... and many other, too many to list... avoid using nondirectional terms (modulates, regulates, etc: in which direction? inhibition or activation? "abnormal" ??? expression what is normal and is it going up or down?) and many other, with the worse being 80% "fusion", instead of "confluence". Acronyms need definition the first time they are used.
Reviewer 3 Report
Comments and Suggestions for Authors
Zhou et al. revealed a molecular mechanism of calycosin-mediated smc calcification suppression. Overall, the experiments were done properly and the manuscript has been well-written. There are several suggestions for moderate revision.
1. Authors need to use primary smooth muscle cells for parts of their experiments to support the results obtained using A7r5 cells.
2. In figure 6c, VAMP8 knockdown doesn’t seem to work well and authors need to use better siRNAs.
3. In page 1 line 37 OPN is abbreviation for osteopontin not osteoblastin?
4. In general, some English editing is required.
Comments on the Quality of English LanguageModerate English editing is required especially for the text (abstract is good).
